# Teacher Training to Take Care of Students at Risk of Exclusion

**María Trinidad Cutanda-López** [1,*] **and María Begoña Alfageme-González** [2]

1    Department of Theory and History of Education, University of Murcia, 30100 Murcia, Spain
2    Department of Didactics and School Organization, University of Murcia, 30100 Murcia, Spain
*    Correspondence: lopez.cutanda@um.es

**Abstract:** We present an overview of how teacher training can work as a key element in good professional performance with students at risk of exclusion. The work derives from a doctoral thesis that analyzes a school reengagement program in the region of Murcia (Spain): Occupational Classrooms. The research was theoretically based on an ecological approach of risk of educational exclusion and a multidimensional approach of school engagement. Focusing on a mixed-methods approach with a multilevel convergent nested design, it was possible to investigate multiple levels: macro (policies), meso (school), and micro (classroom), as well as interrelated elements that influence in the possibilities of re-engagement of these students. The results showed notable deficiencies in the professional performance of teachers working with students at risk, revealing difficulties in the professional development of teachers linked to administrative, institutional, cultural, and personal determinants. Similarly, negative repercussions were detected stemming from the involvement of the teaching staff and on the results of the students and their options for continuity in their educational trajectory. To conclude, the importance of cultivating both the necessary conditions for adequate training, as well as those aimed at making teachers feel supported, included, and recognized, were highlighted in this study.

**Keywords:** re-engagement programmes; students at risk of exclusion; teacher professional development

## 1. Theoretical Framework

The goal of achieving educational systems that guarantee fair, equitable, and quality education has been a central and very current concern on the international political agenda due to its far-reaching repercussions for the individual and society in general. However, there is also unanimous concern about educational aspects that require an urgent improvement for this purpose. Undoubtedly, among these aspects are the low graduation rates in compulsory secondary education and early school leaving (AET, acronym in Spanish), issues that continue to leave many young people out of the system today.

In the context of the 2030 Agenda and in accordance with the framework of Education for All, UNESCO warns of the danger of not paying the necessary attention to students who are in school, but who are not learning: it is necessary to ensure that all boys and girls complete primary and secondary education, which must be free, equitable, and of quality, and must be aimed at obtaining relevant and effective learning outcomes (UNESCO 2015).

In Europe, and after the difficulties in lowering the AET rate to the 10% established for the Union as a whole in 2020 (15% in Spain), the new strategy for 2025 includes among its initiatives the so-called procedures for school success. These aim to ensure that all students can achieve and maintain mastery of the basic skills that allow them to participate fully and actively in society and in the labour market. It is therefore a necessary goal to reduce AET, which fundamentally involves young people from disadvantaged backgrounds, and possibly more so in the current times of pandemic.

Along the same lines, the European Council (2021) indicates that students leaving education early continues to be a challenge since it exposes young people and adults with fewer socio-economic opportunities. The report notes that students from disadvantaged

backgrounds are over-represented in early leaving across Europe, and that the pandemic has highlighted even more clearly the importance of equity and inclusion in education and training.

In Spain, the political strategies in response to AET—with rates that are amongst the highest in Europe (16%, according to the Spanish National Institute of Statistics in 2020)—have been notable and constitute an attempt to comply with the objectives established by the recommendations of Europe and UNESCO. Thus, the reduction in AET has been a priority in the latest educational laws, including the most recent reform (LOMLOE 2020). Additionally, taking inclusive and equitable policies as its backbone, the law appeals for improvement in the dynamics of schools and teachers, and to the responsibility of the administration to eliminate the barriers that limit the access, presence, participation, and learning of students that experience socio-educational and cultural vulnerabilities.

Placing in such a scenario the problem of students at risk of dropping out of school, whom we therefore understand to be at risk of educational and social exclusion (Escudero et al. 2013), there are two aspects that should be specified within this context:

(1) The tendency, in particular in the Spanish context, is to refer these students to specific re-engagement programs and measures that are not always designed and implemented according to fully inclusive parameters, resulting in what can be regarded as parallel, stigmatized and segregated pathways (Escudero and Martínez 2012; Marhuenda and García 2017; Tarabini et al. 2015). Furthermore, the consolidation of these programs has been highlighted as naturalized institutional devices in schools that establish structural modifications and hierarchies according to the profiles and trajectories of the students (Rujas 2017).

(2) The importance of teachers, their training, their commitment, and their involvement in the development of their work with all their students. In this regard, there have been persistent warnings from the political, academic, and research spheres (Arnáiz-Sánchez et al. 2021; Bolívar et al. 2014; Escudero 2017; Navarro-Montaño et al. 2021; OECD 2018; OIE/UNESCO 2015; TALIS 2014). By common agreement, the importance of teachers and the reasons for offering a fair and quality education are highlighted: perhaps there is no other expression in the field of education that is more popular and dominant than the one that corresponds to the idea that quality teaching is the key to good schools (Lieberman, in Day and Gu 2015, p. 139). The problems facing the profession, the need to re-professionalize teachers and the constant challenges they currently face (lack of social recognition, preparation, collaboration, great mobility, instability . . . ) are transforming the teaching practice into an increasingly broad and challenging role, especially for teachers working with students at risk of exclusion (Comisión Europea/EACEA/Eurydice 2015; European Commission/EACEA/Eurydice 2021). This particular teaching role has reached its highest intensity as a result of COVID 19.

In addition to the multiple difficulties involved in the methodological and curricular updates necessary in any process of pedagogical renewal, the urgency of reconversion has been added to respond to the new conditions derived from the context of the pandemic. We cannot ignore many other cultural, evaluative, and relational sensitivities that have acquired a special significance among teachers (Budginaitė et al. 2016). The lack of teacher training, and consequently, the great difficulties they face in carrying out their work, particularly in programs aimed at students at risk of exclusion, imply in many cases a decrease in their commitment and institutional involvement, which leads to the decoupling of teaching and the profession (Escudero et al. 2013; González and Cutanda 2017; Timperley and Alton-Lee 2008).

There are multiple factors and conditions that currently hinder the teachers professional development the general lack of training which focuses on diversity; the scant interest of teachers in training dynamics linked to the most vulnerable groups; a training more focused on theoretically understanding the problems that generate the disengagement of students than practical actions on solved them; and the inconclusive contributions on the incidence of the training carried out in this sense in their own and the students' learning. (Budginaitė et al. 2016; Caena 2011, 2013; Escudero 2017; Hanson-Peterson 2013; Gregory et al. 2013). In this regard, there are a significant number of authors that lo-

cate part of the origin of these problems in the neoliberal discourse (Anderson and Herr 2015; Escudero et al. 2013). From the most critical sector, it points to an ideal under which a managerial logic is consolidated and a hegemonic pattern emerges more aligned with a technical professional model than with one that has to integrate theoretical references (not only rhetoric) with practical decisions in line with the parameters of equity, justice and inclusion. In short, they allude to a preeminent model that weakens the fair distribution of professional development in the face of safeguarding the common good (Escudero et al. 2013). Or, as they themselves point out, only from a formative perspective, assumed and committed to by all, is it is possible to train teachers in the face of vulnerability and exclusion.

On the other hand, the progressive disengagement or lack of involvement and commitment of teachers (with direct repercussions on low-quality teaching and situations of inequity) contributes, among other things, to isolation and stigmatization, the loss of personal control, and transition difficulties between educational levels or in the establishment of collaborative dynamics (Crosswell 2006; Esteve 2010; Moreno 2006; Morriana and Herruzo 2004). To reverse such a situation, specialists in the subject affirm that it is necessary to go beyond an eminently psychological approach that has been predominant (the burnout teachers syndrome) focused on motivational factors. (Salanova et al. 2000). That is, it is necessary to consider other cognitive components (knowledge, capacity and professional development and emotional aspects (teaching vocation) that affect the disengagement of teacher. In this last sense, it can be stated that professional development is in itself a protective factor against disengagement (Ingvarson et al. 2005; Høigaard et al. 2012; Meister 2010; Veláz de Medrano and Vaillant 2010).

In short, the training and involvement of teachers are essential and must be guaranteed from the perspective of the right to education that corresponds to all students. Each student not only has the right to connect with educational opportunities, whatever the motivation they may have and the knowledge or skills they possess. A right that is difficult to guarantee without teachers who maintain their commitment to the constant motivation of their students toward learning. And therefore, teachers committed and passionate about their own learning (Day and Gu 2015).

The training and involvement of teachers are essential requirements for an educational system to function and be able to achieve the progress of all its students, as well as the reaching of their maximum potential. In the face of shortcomings such as those mentioned, the recommendations of the European council or reports such as TALIS are oriented towards new trajectories of improvement, from policies and programs for the development of action frameworks around teacher progression to the strengthening of teacher involvement and commitment. Finally, and among the so-called effective teaching policies already pointed out by the OECD in its 2018 report, the focus is placed, in addition to legislation, on the importance of two aspects of schools: (a) facilitating and designing opportunities for professional development(workshops, or evaluation mechanisms for continuous improvement); and (b) granting greater autonomy for schools themselves in the selection and hiring of teachers. In this way, schools become key agents not only to promote teacher professional development but, in turn, to protect against factors leading to teacher disengagement.

Under the exposed background, the purpose of the article is to show an overview of what are the perceptions of teachers regarding their training and involvement (engagement) when developing their work with students in compulsory school age in situations of severe risk of social and educational exclusion. Specifically, we aim to the following questions: What are the institutional and administrative coordinates under which that condition the selection of teachers assigned to re-engagement programs. What is the teacher profile (experience-training) assigned to these programs. To what extent are teachers involved in their own professional development What is the teaching and learning experience they have to offer to their students. What repercussions could be derived from initial teaching training?

The results and conclusions that are presented derive from a doctoral thesis. A school and socio-labour reengagement program in the Murcia Region (Spain) was analyzed.: Occuptional Classroom. The incidence of political, institutional, curricular and organizational aspect in the results of this program (achieved learning and re-engagement of their students) was studied. This is a program aimed at 15-year-old students with a long history of school failure, absenteeism, disruptive behaviour, who are strongly disengaged from their education and training and are at serious risk of dropping out. The research was theoretically based on the ecological model of school risk (Escudero 2005), and on the multidimensional perspective of school engagement (Fredricks et al. 2004).

The data were collected through questionnaires, interviews and classroom observations of the teachers involved in the seven Occupational Classrooms implemented during their experimental phase in the 2016–2017 school year.

The interesting conclusions drawn concern educational policies but also the schools, teachers, families, and the different educational and social agents, who face a reality of school vulnerability that is as present as it is worryingly invisible in many cases. The findings achieved contribute to the outlining of this reality by showing first-hand the perceptions and difficulties faced by the participating teachers, with far from desirable training and involvement. In this way, personalized and first-hand knowledge contributes to the review and to updating the subject addressed in the face of new proposals for remodelling frameworks and trajectories for training, and for the cultivation of conditions capable of encouraging and maintaining teacher involvement, which is the key in reversing the processes of student disengagement. Some conclusions are drawn that, in short and in general, are intended to contribute to arousing the interest of all those who share an educational perspective in terms of equity and inclusion.

Finally, what is presented constitutes a starting point for the ongoing research project focused on professional teacher development, among whose objectives is to explore the interactions that take place in intergenerational professional development experiences, giving importance to collaboration between teachers of different stages, schools, ages, experience and generations for its repercussions on the professional initiation of teachers. The contributions of the article are aligned with it insofar as they allow for the contextualization of the need for such collaboration as a transcendent support, especially among new teachers and with students in vulnerable situations.

## 2. Research Methodology

This section aims to present in the first place the approach, research design and objectives of the study whose results and conclusions are of interest for this article. Next, the participating population is addressed, followed by the data collection and analysis procedure. The section closes with the pertinent allusion to the criteria that guarantee the rigour of the investigation.

### 2.1. Research Approach, Design, and Objectives

The research, from which the results that are analyzed and presented here are extracted, was positioned in a mixed research methods approach in accordance with the complexity of the problem of school risk that requires a complex analysis from multiple levels of analysis (Cutanda 2021). Additionally, we proceeded according to previous research focused on the analysis of training and the commitment and involvement of teachers in general (Teacher Engagement), and more specifically, on re-engagement programs (Cutanda 2021; Day et al. 2013; Gasiewski et al. 2012; Guthrie et al. 2000; González and Cutanda 2015; Scheilder 2012; Sosu et al. 2008).

The research design was a *Dominant Qualitative Method Multilevel Convergent Nested Design*, and it was carried out in a single phase guided by the predominant qualitative method (the one with the least weight, the quantitative one is nested or inserted in the first one) and carried out at different levels or analysis groups. Although, to illuminate the data collected of a different nature and strengthen the study, the data are integrated into the phases of information

collection and interpretation, in correspondence with the more general perspective of the concurrent design (Creswell 2014; De Liste 2011; Hernández-Sampieri et al. 2014; Morse and Niehaus 2009; Kanga et al. 2015; Tashakkori and Teddlie 2010).

The article, therefore, focuses on two objectives that try to answer the questions raised above: (a) analyze the training, commitment and involvement of the occupational classroom teachers with their work and their students; (b) explore the possible repercussions derived from the conditions of said training and its implication for the professional initiation of teaching staff.

### 2.2. Research

All the selected participants belong to the same setting, which has that was fully addressed. That is, the seven Occupational Classrooms in the context of the study represent a finite population or group in which the number of units that comprise it is known, and there is a documentary record of said units. From the statistical point of view, this constitutes less than one hundred thousand units (Arias 2012). The population coincides with the sample, and so it has not been necessary to carry out any sample selection process.

As far as this is concerned, the sample corresponding to the teaching staff in the case of the quantitative data is comparable to the population (Table 1): all the teaching staff involved in the seven Occupational Classrooms studied (26 teachers). Among them, the distribution by age ranges between 28–59 years, with the average being 40 and a half years. Regarding gender, the distribution is practically equal, with male being slightly surpassed by female teachers (53.8% compared to 46.2% of male teachers).

**Table 1.** Population/sample of participating teachers.

| N | Gender | Professional Features | Qualitative Data Collection | Module/Scope |
|---|---|---|---|---|
| S1 | F | Engineer, 8 years of experience | Interview | MP/MO |
| S2 | F | Graduate, 2 years of experience | | MNP |
| S3 | F | Graduate, 9 years of experience | | MNP/MO |
| S4 | M | Engineer, 8 years of experience | Interview/Observation | MP |
| S5 | M | Engineer, 11 years of experience | Interview/Observation | MNP/MO |
| S6 | M | Graduate, year of experience | | MO |
| S7 | F | Graduate, 5 years of experience | | MNP |
| S8 | F | Graduate, 8 years of experience | Interview/Observation | MNP |
| S9 | M | Engineer, 3 years of experience | Interview/Observation | MP |
| S10 | F | Graduate, 13 years of experience | Interview/Observation | MNP |
| S11 | F | Graduate, 5 years of experience | Interview | MNP |
| S12 | F | Engineer, PhD, 9 years of experience | Interview/Observation | MP/MO |
| S13 | F | Engineer, 8 years of experience | Interview/Observation | MNP MO |
| S14 | F | Graduate, 2 years of experience | Interview/Observation | MNP |
| S15 | M | Technical specialist, 13 years of experience | Interview | MP |
| S16 | F | Graduate, 7 years of experience | Interview | MO |
| S17 | F | Graduate, 4 years of experience | Interview | MNP |
| S18 | M | Technical architect, 25 years of experience | | MNP/MO Tecnología |
| S19 | M | Diplomat and technical specialist, 12 years of experience | Interview | MP |
| S20 | M | Graduate, 9 years of experience | Interview/Observation | MNP |
| S21 | F | Graduate, 10 years of experience | Interview/Observation | MNP |
| S22 | M | Engineer y technical specialist, 6 years of experience | Interview/Observation | MP |
| S23 | M | Graduate, 4 years of experience | | MNP |
| S24 | M | Engineer, 9 years of experience | | MNP |
| S25 | F | Graduate, 10 years of experience | | MNP |
| S26 | M | Graduate, 7 years of experience | | MO |

Source. Own elaboration.

The professional profile is basically that of graduates or engineers, practically all of them with experience in Compulsory Secondary Education (ESO, in Spanish acronym) (57.7%), as well as in other fields of education, among which the different vocational training profiles stand out (42.3%). Only four teachers lacked experience in measures of attention to diversity, basic vocational training or adult education. However, if we take into account the previous professional background in the program studied, only five of them have experience in the Occupational Classroom (S6, S12, S15, S19, S22), four of whom are tutors. Therefore, more than 80% of the teachers had not had direct contact with the program until the moment of their incorporation.

Finally, and in terms of employment status, the vast majority are in an interim situation and are therefore engaged in provisional employment (92.6%) compared to the rest (only two teachers), who have achieved civil servant status (S6, S18). Table 1 is complemented by the professional training module or area taught by teachers, which we have distinguished between professional module (MP), non-professional module (NPM) and optional module. In addition, regarding the qualitative information, the data of the subjects who have participated in the data collection (interview or observation) appear. A total of 17 teachers from all the programs studied were interviewed. Additionally, it was possible to access five of the seven Occupational Classrooms analyzed, in which a total of 11 teachers could be observed.

### 2.3. Methodology of Collecting and Analyzing Research Information

The qualitative and quantitative data related to the teaching staff were collected in a single phase once all the pertinent permits had been obtained: from the Regional Administration; Secondary Education Centres in charge of the programs; Occupational Classrooms and with the explicit consent of the teachers. The main considerations on the instruments for collecting information and its analysis are set out below, taking into account the nature of the information in a differentiated manner.

The quantitative data were collected through an ad hoc questionnaire based on theoretical research references and taking into account previous research instruments (Amores 2013; Escudero et al. 2013; Escudero 2017). The questionnaire was structured in VI large dimensions (professional/identification data; students; teaching in the Occupational Classroom; the secondary school; the Occupational Classroom; and training and professional development), diverse variables (whose operationalization can be seen in Table 2), and a total of 28 mostly closed questions, with four response options, and ordered according to the funnel technique (Hernández-Sampieri et al. 2014, p. 228). The questionnaire was validated by expert judgment or the individual aggregate method, according to the specifications of Corral (2009). Its validity was guaranteed (theoretical foundation, consultation with experts and application of the measurement instrument); its reliability was assured with a Crombach Alpha value of 0.876, an adequate value according to George and Mallery (2003) and Gliem and Gliem (2003); and all 104 applied items and their objectivity were verified through standardization when applying the instrument and through theoretical–practical personal training. All the participating teachers completed the questionnaire, distributing the sample among the teachers of the seven Occupational Classrooms.

Quantitative data were analyzed using univariate descriptive statistics with the support of SPSS 24 Software. In particular, through the following statistics were used: (a) measures of central tendency (mean, median and mode); (b) measures of variability or dispersion (range, standard deviation and variance), and (c) studies of data distribution in terms of probability and dispersion (asymmetry and kurtosis). The qualitative variables of the questionnaire were analyzed using frequency distributions and percentages. The quantitative information made it possible to obtain a first panorama of the teaching staff at an exploratory level that was deepened with the study of the socio-educational reality through the analysis of qualitative data.

Qualitative data were obtained with two instruments: the interview and observation in the classroom. The data presented here correspond to semi-structured or guided,

confidential and authorized in-depth interviews. Their structures followed the logical order recommended by McMillan and Shumacher (2005), grouping the questions by major topics of interest (roles and functions linked to the program; ascription criteria, training and involvement in the program; relationships, evaluations of the program). A total of 17 interviews were conducted. The information was analyzed using a deductive–inductive content analysis model (Cáceres 2003) with the support of the Atlas ti V8 software, in the following phases: 1. pre-analysis (information organization and material review; decisions: cases, technical and strategies; memos); 2. definition of analysis units; 3. open coding (analysis rules and classification codes); 4. definition of categories.

**Table 2.** Operationalization of variables of the questionnaire aimed at participating teachers.

| Variable | Dimension | Indicators |
|---|---|---|
| Personal Characteristics | Personal Profile | Sex; Age |
| | Professional Profile | Academic degree; specialty; professional experience; employment situation; incorporation to the program |
| Organizational conditions | Structural | Role; membership Organizational Unit; formal work and coordination mechanisms; structure of the tasks; physic structure |
| | Relational | Formal; informal |
| | Procedural | Planning and strategies; evaluation improvement and innovation; management and leadership; training. |
| | Cultural | Rules; values; assumptions |
| | Environment | Mediate; immediate |
| Educational dimensions | Curriculum | Design; monitoring and evaluation |
| | Teaching–learning process | Objectives and principles; contents; methodologies; relational climate; Mentoring and guidance |
| | Evaluation | Tracing; evaluation criteria and methods |
| Teacher Engagement | Stall | Cognitive engagement (teacher training) Emotional engagement |
| | Contextual | Social engagement: relationships; perceived social support; perception about personal relevance; political considerations |
| Teacher professional development | Participation | Contents; activities |
| | Assessment | Utility; supports/facilities |
| | Incidence | Teaching learning; student learning |

Source. Own elaboration.

### 2.4. Research Rigour

The rigour of the research was guaranteed based on: (1) design quality (adequacy, internal consistency, and faithful implementation in practice); and (2) interpretative rigour or quality inferences: coherence with respect to the theory from start and transfer among classrooms—non-generalized—in accordance with the dominance of the qualitative approach, as well as the particularity of the studied programme given that it is carried out as a pilot programme in a single Spanish autonomous community and all the Occupational Classrooms implemented during its experimental phase (2016–2017 academic year) were studied (internal–external).

### 3. Results and Discussion

The two stated objectives that are answered in the light of the research results are the ones that will guide the exposition, argumentation and discussion along the following lines. Nevertheless, and to situate the reader, a reference will be made beforehand to provide a brief normative characterization of the programme addressed.

### 3.1. Normative Characterization of the Occupational Classroom

The main and defining characteristics of the programme studied according to the legislation in force at the time of its analysis are summarized in Table 3.

**Table 3.** Purpose and main characteristics of the Occupational Classrooms in the Region of Murcia.

| | |
|---|---|
| Purpose | Reduce absenteeism and the risk of early school leaving by providing students with the personal, social and professional skills that favor:<br>(a) their continuity in the education system, preferably in a Basic Vocational Training training cycle or in a Vocational Training Program—adapted mode.<br>(b) Their socio-labour insertion and their incorporation into active life with responsibility and autonomy. |
| Recipients | Students who are 15 years old and who must take the 2nd or 3rd year of ESO during their year of access, who are not in a position to continue in higher levels, have repeated an ESO course, and have an open absenteeism file. Students with a curricular gap in the block of core subjects and behaviours contrary to the rules of coexistence in the centre during the course prior to their access are prioritized. |
| Structure Modular | (a) Modules associated with units of professional competence.<br>(b) Modules and general fields not associated with units of professional competence: Field of Applied Sciences. Sociolinguistic field. Two optional modules.<br>(c) Work Centre Training Module. |
| Ratio | 8–12 students (in one group) |
| Duration | A school year |
| Certification | Academic certification of modules and areas studied and accredited units of professional competence. Overcoming the latter will be cumulative for recognition in subsequent studies. |
| Title | Does not title |

Source: Own elaboration, according to a Resolution for which instructions are issued for the adaptation, on an experimental basis in the 2016–2017 academic year, of the occupational classrooms for the 2015–2016 academic year (Resolución de 27 de julio de 2016, de la Dirección General de Innovación Educativa y Atención a la Diversidad).

According to this table, the Occupational Classrooms are a programme designed to provide academic training (linked to subjects taught in ESO, although in a more flexible way -in modules) and have basic professional training that allows the re-engagement of students on a school trajectory characterized by negative behaviours, absenteeism and school disengagement. It takes place in municipal facilities, outside the secondary schools, although administratively linked to one of them.

Once this measure has ended and has been successfully completed, returning to ordinary ESO is not prioritized: students can preferably continue studying a Basic Vocational Training programme if they meet the access requirements, or an adapted "version" of this Program (Professional Training Program—Adapted) that is usually implemented in non-profit associations or NGOs.

*3.2. Training and Involvement of the Occupational Classroom Teachers with Their Work and Their Students*

Before fully discussing the results related to teacher training, it is worth briefly mentioning the conditions and predisposition of the participating teachers prior to joining the Occupational Classroom.

From the analysis of the questionnaire responses to items in which the teachers were asked the reasons why they accessed the programme in question, the results bifurcate between the interest in teaching in this modality (51.8%), a perception that corresponds practically in its entirety to the tutors of said Classrooms and that is nuanced in the course of the interviews, and other reasons that are quite different from personal motivation. Among the latter, the predominant reason given is that they had no other choice (37%) followed by proximity to home (11.1%), the need to complete the working day (7.4%) and in last place was the fact it was a substitution position (3.7%) or because it was awarded by the administration among the compensatory teaching staff options (3.7%).

If we take into account the qualitative information collected through the interviews, we can say that the formal process of assigning the teaching staff to the programme was carried out in two ways: (a) by adjudication in a competition between the vacancies in the compensatory (among which is the Occupational Classroom) (b) at the proposal of

the management team among the staff of the school with available hours, in the case of Non-Professional Modules. Although the ascription is voluntary and requires informed consent, in the interviews there is a greater reiteration around the perception that in a certain sense it was "forced", and even comes to be seen as a "punishment" by these teachers who are to accept due to their unstable job situation.

> *When you apply, you can read Secondary, it doesn't say anything about the Occupational Class . . . As there were four options and I couldn't reach for the others, it's the only thing you can pick and it was the last choice, so I had it assigned to me. However, I didn't know anything, not even that the Occupational Classroom existed, which is for children who are outside the normal sphere. It's even like a trap . . . you accumulate experience that you need and, being an urgency, you take it, the classes that are given here are something different, and it's not for everyone. (S11)*

> *There are people who do not know what the Occupational Classroom is and those who know, do not apply for them. (S19)*

> *I am on a list of interns specialized in Vocational Training in one of the last positions, and people generally do not choose Occupational Classrooms because they do not want to teach in an Occupational Classroom. It was the last award and they assigned it to me, in our case, we take what nobody wants. (S4)*

> *Interim teachers work in list order, when they list the available positions, in that moment it's up to you to choose from what there is available and they offer you, that's what you pick. There when a substitution or an Occupational Classroom position is available, it is not usually advertised as such. At the most, it is listed as "compensatory", but it means support classes in general, you never know what you are going for, most specially when there is one of these specific programmes. (S10)*

It is also noteworthy that many of them were informed at the last moment of their assignment to the Occupational Classroom and had little or no information about its operation until well into the course: *At the secondary school they did not explain to me either[ . . . ] there are things that maybe they should have told me, but they didn't do so because of the work pace [ . . . ] However, of course I'll get there and manage* (S4).

These were conditions that, with few exceptions, mostly led to feelings of surprise, uncertainty and rejection of teaching in the programme. This follows from statements such as the following: *When you arrive and find what you find, you say, My God, what is this! Where have I got myself into? (S15); We knew it was going to be hard when we started seeing the students (S20).* These perceptions are aggravated, if possible, by the stigma attached to the programme and ultimately generate a negative bias with significant repercussions for teachers: *Teachers with more experience told me to be careful! (S11). Some people don't know what the Occupational Classroom is [ . . . ] but if some people hadn't been involved, we wouldn't have had any people at the end of the course (S15).*

Based on these approaches, it can be argued that the criteria for the access of teachers to the programme are subordinated, both by the administration and by the centres, to the hierarchy under which the status quo of the school order is sustained. Consequently, the response to students at risk of exclusion must be provided by the last ones to arrive, who are the least prepared and who have with less experience in correspondence with their lower status. This constitutes an issue that not only overlooks the difficulties that the exercise of teaching work in the programme can entail, but also contributes to a considerable decrease in the degree of involvement teachers can have with their students. Additionally, for what is concerned here, there is also a decrease in the necessary commitment to improve their ability to respond to the needs of these students. Therefore, in the possibilities that these students can reengage with their educational and formative trajectory (Williams, in Guthrie et al. 2000). On the contrary, the profile of these teachers poses a scenario highly likely to encourage their own disengagement, and with it the increased probability that the teaching offered will be of low quality (Giangreco et al. 2001; Crosswell 2006; Esteve 2010; Moreno 2006; Morriana and Herruzo 2004) among other issues, due to the difficulties

that this work entails when implementing necessary renovations and changes (Amores and Ritacco 2017; Araméndi et al. 2011; Barbero et al. 2018; Fernández-Enguita 2011; Rodríguez-Rabadán and Candela 2016).

The following sections, which address the relative results of both teacher training and their perceptions of their teacher training, point in that direction.

3.2.1. Occupational Classroom Teacher Training

As indicated when addressing the sample of participating teachers, their initial training corresponds to various areas of specialization randomly based on the ambiguity and uncertainty that could be attributed to the ascription process. With few exceptions, there is no specific training required to work with students at risk of exclusion who attend programmes such as the one studied, nor, for the most part, is previous experience required in them. Such conditions seem not to have been taken into account if we consider in what terms the professional development of participating teachers is deployed. Only four of the 26 teachers (15.4%) were involved in some training dynamic related to the attention to students in vulnerable situations. Two of them also argue that they have done this training to add professional merit, and only three declare that they took advantage of it being useful and helpful for their students. The vast majority of teachers (81.48%) attribute such training deficiencies to the lack of courses offered by the administration, and, to a lesser extent, they alluded to the lack of time available to get involved in training dynamics. Finally, it is noteworthy that the majority allude to the fact that training expressly aimed at improving the teaching–learning processes of the most vulnerable students ia not among their priorities. For their part, they tend to attach importance to a more technical training, which they consider to be adequate in relation to the profile and teaching function, which many of them think is essentially focused on the content of students in ordinary classrooms.

> *Teachers are not trained because a single person cannot know everything. If I dedicate myself continuously every year to teaching in an Occupational Classroom, I train for Occupational Classroom. However, if, for example, this year I am in Technology, I have taken three, four courses; last year I was in Building and civil works, I took three, four courses; Another year I was in Electronic Systems, I took Secondary Education courses. And one is not God, then . . . but it is that even a professor with a position is also constantly jumping and changing . . . One is trained in what he is interested in because of course maybe next year I will no longer teach that subject. (S5)*

Based on the previous evaluations, we can say that it is once again laid on the table that teacher training is an element as important as it is neglected, particularly regarding the attention of the most vulnerable students and at a triple level:

(1) The lack of prominence that the administration gives to those subjects and training modalities that are essential to addressing the problems of failure, absenteeism, disengagement and school dropout, whose lack of supply is highlighted by the teachers participating in the study. The results indicate that teachers follow the line set by training policies and that their preferences tend to opt for what they consider to be the highest priority. Precisely, they prefer content training aimed at providing expert knowledge and not one designed to help train a less technical and more reflective professional. All this encourages a serious reflection on teacher training policies, particularly in secondary schools.

(2) Secondly, this is due to the schools, where the lack of time and space for training seems to continue to be the general trend.

(3) In the third and last place, this is due to the teachers themselves, for whom the importance given to said training is scarce. In the case of opting for some types of training, their situation of professional instability leads them to one that can provide them with greater expert qualification to expand their interim options, or to increase their professional merits in the face of exams for public service. From this position, indifference or passivity in the face of greater preparation to serve students at risk has been notorious.

> *No, no, the truth is I have been able to separate very well the Occupational Classroom and work at the school. Each field has its characteristics . . . so maybe, maybe you do*

*extrapolate one thing to the other, the only thing that can help me is that I can reuse some type of material that I have given them (the students at the Occupational Classroom) that been of use to me with some very weak students from the first year of ESO, but [ . . . ] not for classroom management because it is very different, it is all very different, it has nothing to do with it. And methodologically speaking neither, they are very different methodologies. (S10)*

A far from negligible approach for considering the difficulties in attending to the increasingly diverse student body that accesses Secondary Education in any ordinary group, as confirmed by research (Budginaitė et al. 2016; Caena 2011, 2013), and as has even been emphasized by the interviewed teachers themselves, is outlined:

*In the school now, they have a big problem because they have a couple of ESO courses in which the first levels are horrible in behaviour . . . and it's getting worse, eh . . . (S20)*

*In all the schools there are classrooms and classrooms. I have come across many classrooms that have also been like these and were public and concerted secondary schools. Last year I was in a school where I have seen blows, shaving to separate them . . . there were kids who were even worse, more conflictive. (S11)*

Therefore, conclusions are drawn that coincide with and highlight what has already been put forward in previous works, at an international nature when addressing teacher training in general (Hanson-Peterson 2013), at the national level (Escudero 2017) and or in programmes and measures of attention to diversity, and as shown by other studies in the same study context (Arnáiz-Sánchez et al. 2021; González and Cutanda 2015, 2020b).

3.2.2. Teacher Training: A Biased View

The general lack of training indicated in the preceding section, however, contrasts with the perception regarding their own training (Table 4). When the participating teachers were asked in the questionnaire about the "training needs" of teachers in general, the answers were unanimous in pointing out training deficiencies in practically all the aspects raised. There were only specific exceptions regarding sufficient training on evaluation issues (14.8%), curricular adaptations (11.1%) and joint reflection by the teaching team (7.8%). However, and from a very different approach, it is noteworthy that the answers about their own training needs showed a much more positive self-perception. Teachers consider themselves sufficiently trained in: (a) evaluation aspects (33.3%); (b) pedagogical possibilities of information and communication technologies (29.6%); (c) strategies to manage student demotivation, individualized curricular adaptations, and work with families (25.9%); (d) characteristics of the students and methodologies (22.2%); (e) classroom management and reflection in teaching teams (18.5%); and (f) analysis and reflection on the teaching itself in order to improve it.

It is also possible to specify that, for a good portion of these teachers, such training can be replaced with experience. This is, many participants put experience before the need for training as the main source of training, despite the fact that for most it is their first foray into the programme.

*All the training is logically good; however, I think that what has saved me the most has been the experience (S10)*

*I think that nowadays, after all, what shapes you is experience (S11)*

*It is the experience of personal life, the theoretical experience in these chaos does not give you everything either or what you have gone have gone through an internship . . . . (S22)*

Such statements contrast, however, with the perception that becomes evident when the interviews on the daily evolution of teaching in the programme are deepened, and in which the lack of training comes to light in an evident way:

*I should have (specific training) because I am an engineer. (S12)*

*I act according to my intuition and sometimes I lack training, I lack presence, I lack things to know how to act (S19)*

*There should be some kind of training for the teachers that come here regarding the students that we deal with (S13)*

*In order to get more out of these students, you have to be highly trained ... there is a lack of training for teachers who are not specialists. (S5)*

**Table 4.** Degree of agreement of the teaching staff on the training necessary to work with students at risk, and on the aspects on which they would need more training in particular.

| Categories | Training Needs | | | | "Their" Needs | | | |
|---|---|---|---|---|---|---|---|---|
| | NA | DIS | A | TA | NA | DIS | A | TA |
| 28.1. Characteristics of the student and its importance for teaching–learning | 0% | (1) 3.7% | (8) 29.6% | (18) 66.7% | 0% | (6) 22.2% | (12) 44.4% | (8) 29.6% |
| 28.2. Climate and classroom management coexistence and conflict resolution | 0% | 81) 3.7% | (11) 40.7% | (15) 55.6% | 0% | (5) 18.5% | (12) 44.4% | (9) 33.3% |
| 28.3. Techniques and methodological strategies to respond to diversity | 0% | (1) 3.7% | (11) 40.7% | (15) 55.6% | (1) 3.7% | (5) 18.5% | (9) 33.3% | (11) 40.7% |
| 28.4. Strategies to manage student frustration and demotivation | 0% | (1) 3.7% | (7) 25.9% | (19) 70.4% | 0% | (7) 25.9% | (6) 22.2% | (13) 48.1% |
| 28.5. Pedagogical possibilities and didactic applications of ICT | 0% | (1) 3.7% | (15) 55.6% | (11) 40.7% | (2) 7.4% | (6) 22.2% | (13) 48.1% | (5) 18.5% |
| 28.6. Individualized curricular adaptations | 0% | (3) 11.1% | (9) 33.3% | (15) 55.6% | (1) 3.7% | (6) 22.2% | (15) 55.6% | (4) 14.8% |
| 28.7. Criteria and procedures for student evaluation | 0% | (4) 14.8% | (14) 51.9% | (9) 33.3% | (1) 3.7% | (8) 29.6% | (14) 51.9% | (3) 11.1% |
| 28.8. Analysis of and reflection on the teaching itself to improve it | 0% | (1) 3.7% | (12) 44.4% | (14) 51,9% | (1) 3.7% | (3) 11.1% | (16) 59.3% | (6) 22.2% |
| 28.9. Analysis/reflection of the teaching team on the results deciding improvements | 0% | (2) 7.8% | (10) 37% | (15) 55.6% | (1) 3.7% | (4) 14.8% | (15) 55.6% | (6) 22.2% |
| 28.10. Activities with families/environment to improve relationships and participation. | 0% | (1) 3.7% | (16) 59.3% | (10) 37% | 0% | (7) 25.9% | (6) 22.2% | (13) 48.1% |

Legend: NA (no agreement); DIS (disagree); A (agree); TA (totally agree)

Source. Own elaboration.

In some cases, the lack of training is directly correlated with the lack of commitment and involvement of the teaching staff. A matter of far from negligible consequences in terms of its impact on the teachers themselves, many of whom have experienced feelings of impotence, frustration and stress to a greater or lesser extent.

*At first it is true that my idea is always to throw myself into those who are worse off and it is frustrating, and then the social educator told me: "Look, this always happens. Some years we can save but one" (S20)*

*This has happened to the substitute, he had no idea where he was coming to and, of course, at first he was demoralized: "when will I be able to start my teaching?" and I told him: "you don't come here to teach the way you used to understand teaching, here you have to work on other skills, discipline and by working on that, you get them to learn two times four, that is what you and them take away". (S10)*

On the contrary, is the students' own learning in which they place scant expectations and to which they address themselves by focusing on the difficulties that characterizes them. That is, there is evidence supporting a deficit approach, in the terms referred to from the theoretical point of view, which we can see in the two previous quotes and also in these:

*I have these kids with a level of 3rd grade of Primary of not knowing how to multiply (S20)*

*All sorts of curricular level difficulties (S8)*

*We have students here with a level of 5th grade of Primary school whose level is far above them, little more than adding with their fingers, they know nothing, the curricular gap is the most characteristic in this group (S5)*

*The student himself is the one who already brings vices, errors of education so to speak from home that mean that many of them do not go to the classroom (S4)*

*They all have problems, one is because he got his girlfriend pregnant; another is because his father has a junkyard and wants to work there, and others who turn 16 and can fly (S12)*

*+The greatest absenteeism problems tend to be of the Roma ethnic group because they immediately start working on the weekly street markets [ . . . ] and leave (S19)*

*Here they have carelessly been banking the doors, corridors, in the elevator, spitting on the floor . . . (S10)*

*Their way of speaking is aggressive in general, I mean that, although they speak well, but they are threatening, they raise their tone at you, they make you feel uneasy (S11)*

A deficit approach being still very present in the organizational culture of the centres (González and Cutanda 2020a, 2020b), and that is in line with what was pointed out in the previous section, would mean that the aforementioned training need is not considered a priority. Thus, there is an almost commonly accepted view that the teaching staff that has to attend to the particular needs of students in vulnerable situations must be a specialist teaching staff with a permanent assignment to programmes such as the one studied, while the rest of the teachers "disengage from the programme": *At school the teachers know what they will find in those classrooms and some of them looking forwards to see fly some of the students, where? Wherever because they are disruptive (S5).* Additionally, therefore, they have little or no knowledge about the Occupational Classroom, beyond the stigma attached to it:

*Here the profile of the student body that goes to the Occupational Classroom is known, what is done there, and what is achieved, and the final results, no . . . Most of the teachers do not know the required amount of work to be done, and what teachers are suffering and those results at the end (S19)*

This is a situation linked, according to the teachers of the program, to various aspects: (a) the lack of support requested and received among the group of professionals at the centre; (b) his lack of training; and (c) The fact that they consider solving behavioral problems (disruptive behavior and absenteeism problems) a priority over the learning of their students. In addition, it is not uncommon for disparate positions to exist when dealing with such problems. Arguments such as the following confirm this:

*In the face of a problem when at work the one who acts is the Head of Studies (S5)*

*The teachers, when they come I talk to them, but most of the time I am alone (S4)*

*It doesn't exist here, it's a lack, there should be constant coordination meetings, but it takes the extra hours I have as a paperwork management tutor, tutoring, family care . . . and you juggle and organize yourself (S22)*

*Normally we talk about the profile of the students and well, sometimes we have had conversations in which we assume that the teachers themselves are wrong (S4)*

*So, we deal with problems, for example, at the beginning of the year there have been many problems of adaptation to the teachers because they have a different approach, so we have had to adapt to them, study each case and we more or less know what they were doing in Primary, why they were absentees, what conflicts they had . . . and more or less we have been able to guide ourselves (S15)*

*Whenever we have a disruptive behaviour, we have to report it there, and it is discussed during the coordination hours . . . However, 99% of the time the school is not even aware of the disruptive behaviour, we solve it in the meeting that we have us weekly at the assembly. Some attitude of theirs some teacher already excessive, excessive, excessive, then you transfer the student to the school, but as soon as you transfer him, the centres have a regulation (S1)*

As a whole, an image is offered that reflects the lack of training of the teachers of the Occupational Classrooms in a sense similar to that argued in previous investigations in the Spanish context in order to document good practices with vulnerable students. Although, in general terms, these problems start from a lack of training, the need for improvement has been a key trigger for the achievement of good practices. Thus, for example, the results of Amores and Ritacco (2017) confirm the value of involvement from teachers in training dynamics. Along the same lines, Fernández-Batanero (2011) documents the great importance given by teachers to training based on their own reflection. Pareja and Pedrosa (2009) argue that the fact of significantly involving teachers in the development of their own learning is a crucial matter in the face of real options for improvement, generating the necessary changes with respect to deeply rooted conventions, as occurs with this type of teaching programme and the students they receive. In the case of the Occupational Classrooms, as has been verified, it is a feeling and a fact absent in the majority of their teachers, which is in line with what happens in general among secondary education teachers (Araméndi et al. 2011).

The teaching practice in these Classrooms, understood individually and as a responsibility that only concerns their teaching staff, therefore forgets the support, collaboration and exchange of professional knowledge, thereby maintaining the pedagogical status quo that turns these teachers into isolated residents in their helplessness (Gimeno-Sacristán 1998) in the face of the multiple difficulties they have to face practically alone. Consequently, and although we could not openly speak of a rejection of this segment of the teaching staff, neither could it be said that there is full sustained integration based on values of collaboration and support for teachers who are seen as "brave" but also "different". In some cases, there is even talk of professional isolation; particularly among the teachers or the program who do not teach at the IES. Such a situation is not of negligible importance if we consider that the social support perceived in the workplace has a direct impact on the well-being of teachers and on the perception of their personal and professional relevance. Therefore, and according to the exposed panorama, it is possible to speak of a dysfunctional culture since it negatively influences the professional development of teachers.

*3.3. Explore the Possible Repercussions Derived from the Conditions of Said Training and Implication for the Professional Initiation of Teachers*

The results that have just been argued and discussed draw a scenario marked by multiple signs of despondency among the teachers assigned to the Occupational Classroom. A situation linked to their lack of training that the teachers argued is due to administrative, institutional and cultural problems and also to their lack of personal involvement. Thus, a story is constructed with important derivations for the professional initiation of teachers in both directions, which supports and provides feedback to the incipient results of the ongoing research entitled "Intergenerational Professional Development in Education: Implications in the Professional Initiation of Teachers"

This last project adopts the association between centres that provide quality teaching and the existence of a solid and widespread learning "culture" among its students and teachers as its starting theoretical references. Regarding the teaching staff, this is closely

linked to their professional development, something understood not only in relation to quality technical training but, in turn, to the relationships and the perceived support among the participants. However, and starting from such premises that would go in the same line of discernment of the theoretical precedents argued here, the project in question assumes the need for an "integrated professional culture" (Kardos and Johnson 2007), characterized by professional exchange and mutual support between more and less experienced teachers. In this regard, the generational relevance of teachers at different stages of their professional career has been taken into account, among other factors. This is a question that covers aspects such as: what teachers with different experiences understand by professional efficacy, their position in the face of relevant organizational and institutional changes, the use they make of professional development or, even, the conditioning factors of disengagement and abandonment in extremis of their position (Vallejo et al. 2021).

Without delving into or going into a greater level of detail that does not correspond with discussions here, it has been deemed appropriate to provide a few brief outlines of certain results derived from qualitative information that, we insist, is at the dawn of its analysis process. However, these data allow us to guide and conduct the initial exploration in terms of those difficulties pointed out for teachers who are beginning their professional career, and who come to subscribe to what was argued above. Thus, we take into consideration with due caution results that are considered solely for illustrative purposes, but which are by no means conclusive yet, and much less generalizable. It is worth highlighting certain considerations derived from the interviews carried oyt with the teachers of Compulsory Secondary Education and Vocational Training: teachers with little professional experience; experienced and retired.

The discourse in this case is thus marked by an emphasis on the difficulties that teachers with less experience, or even those who start their work in a centre where they have not worked before, generally face. Given the existence of barriers associated with their lesser professional and contextual background, and in the latter respect also, by the extension of understanding the organizational culture in its less explicit aspects, the demand for support for the group of teachers, and particularly the most veteran, is clear. However, the most common response has been a difficulty in providing such support and collaboration in the first place for personal reasons. This is an aspect that is difficult to transform because, for some interviewees, individualism is in the "DNA of the teaching staff in secondary education stage" in the way of thinking that the teacher has today. For some teachers, it even prevails more acutely in the new generations highly influenced by the preponderant neoliberalism, an aspect that enhances individualism and that reduces the options for collaboration, as well as those for their own training and professional development.

Administrative and institutional constraints are defined in this line. Among the latter, the lack of time and space for collaboration, the departmental and balkanized culture existing in secondary schools, or the teaching overload are once again highlighted as the main issues.

In addition, the greatest deficiencies regarding the need for collaboration and support are highlighted by the teachers who work only in these specific programmes and not in ordinary classrooms. A support that they consider essential because their student that characterized by its problematic character. In this sense, it is usual to help a classmate to solve a problem or assist a student, albeit in a timely manner, addressing complex situations individually when these are more common or generalized.

In this last respect, it should be noted that, despite being aware of the greater difficulties that teachers face with the most vulnerable students, a line already addressed in this work is insisted on, since teaching is assigned to the least experienced teachers and in a situation of greater labour instability. The assignment of inexperienced teachers to these programmes is a consequence of the great mobility existing among teachers without a fixed position, which in turn constitutes a hindrance to strengthening possible collaboration ties that are essential to achieving good results with the most vulnerable students. Comments from veteran teachers indicate the need to maintain a stable staff over time and to assign the best

teachers where they are most needed, in this way the final results and teacher collaboration will be favoured.

This is an issue that is also invariably associated with situations of anguish and stress for a large part of the teaching staff assigned to these programmes, as well as to the Basic Vocational Training programmes in general, considering that their work in them much is more complex than that carried out in the ordinary classrooms of Compulsory Secondary Education or the Baccalaureate. Thus, these are statements that are in line with the results derived from the analysis of the Occupational Classroom. It is confirmed that the deficit approach is contrary to the possibilities of school success. It contributes to reduce the effectiveness of the teaching staff insofar as it encourages low expectations of achievement in the case of most vulnerable students. Some teachers point out that success as a social integration of the individual and his human formation should be taken into account before school success.

Finally, it is possible to conclude this brief exposition by referring to particularly relevant aspects in order to overcome such difficulties that have been commented on by the professors of the ongoing research: (a) it is important to welcome newly incorporated professors into the centres; (b) it is important to implement training dynamics in schools based on their own and particular needs detected and the group of students to which a response must be given. However, furthermore, it is important that all the teaching staff of the schools participate under a pedagogical leadership capable of joining efforts towards a common and democratically shared project and, likewise, with the support of the administration.

However, it is also pointed out that for this to be possible, it is first necessary to break with a predominant and strongly established culture, very different from what is understood as "integrated professional culture", with training more adapted to the job position and to teamwork. Secondly, it should be carried out by opening the schools to the social environment in which they are located, especially those schools or programmes that receive the most vulnerable students. Thirdly, it should be carried out involving the most veteran teachers, including retired teachers, who can accompany the new ones with their experience and knowledge. Additionally, fourthly, its implementation should involve the commitment of the management teams themselves in training dynamics from the perspective of pedagogical leadership and promoting the autonomy of the centres.

In light of the considerations arising from the analysis of the aforementioned project, we can say on the one hand that there is firm support for the necessary shift in teacher training, both initial and continuing, for the sake of training that qualifies teachers, especially new generations facing the new and changing panorama of the 21st century, characterized, among others, by the coordinates that emanate from neoliberalism. Both the professionals interviewed from the secondary schools that teach in Compulsory Secondary Education, as well as those of Vocational Training in the national context, agree on this. Additionally, secondly, the training specifically aimed at responding to the needs of students in conditions of greater vulnerability deserves a special mention. Thus, a complex scenario is drawn that can be discussed around the theoretical assumptions that are entrenched in the study of Occupational Classrooms. Among the answers arising from this, without a doubt, is intergenerational teaching collaboration, as Vallejo et al. (2021) highlights is an issue to be addressed to facilitate intergenerational professional development.

## 4. Conclusions

Based on what has been argued in previous sections, the answers to the objectives on which this contribution focuses strengthen the conclusions were already pointed out at the beginning of it: there are verified gaps found in the training of teachers to deal with the group of students in a situation of vulnerability in the desirable terms. These are intimately linked to serious limitations as far as the teacher professional development is concerned. The nature of the difficulties that have been delimited and argued are multiple and disparate and concern all educational agents in the various planes considered in an

interconnected way, which are also in harmony with the preceding theoretical precedents from start and with the complexity of the problem treated. However, and in light of the arguments derived from the study presented here, it is worth detailing the following reflections by way of conclusions from an improvement approach in terms of professional teacher development:

In the first place, it is impossible not to appeal to the convenient reflection on the policies of selection and training of teachers used by the administration. The conditions designed for the access of teachers to programmes specifically aimed at the most vulnerable group of students, in which the suitability of their profile is neglected both in terms of experience and professional training, contributes to the promotion of a generalized discomfort among them, with a consequent decrease in their involvement, among other aspects, with their training process. It should also be added that another key aspect to be considered in the roadmap towards improving professional concerns, without a doubt, concerns the need to rethink the conditions of instability and constant mobility among teachers in general. Although such determining factors suppose a worsening of the already precarious situation of the teaching staff assigned to programmes such as the one studied, these have been verified to be a difficulty that largely concerns the teachers of secondary schools as a whole. This is specifically important, and as far as this is concerned, given the repercussions of this development for carrying out training projects in centers and, more broadly, for promoting and maintaining professional collaboration relationships. This is an issue that, in turn, should be considered in greater detail due to the pejorative consequences that it entails for the professional initiation of less experienced teachers. In that sense, two aspects of improvement which have already been pointed out as far as the political plane is concerned make full sense in this regard. On the one hand, there is a need to facilitate and design opportunities for professional development through workshops, as well as evaluation mechanisms rooted in their teaching practice and aimed at a continuous improvement, as pointed out by the OECD (2018). Additionally, on the other hand, there is a need to shift from more the convenient perspective of change of continuous training that focuses on a technical professional, towards one that allows the full training of teachers as critical and reflective professionals from ethical and social justice approaches (Escudero 2017). This latter topic opens the door to the hackneyed and persistent debate between how to achieve a balance between equity and quality. Additionally, we must rule out training policies influenced by neoliberal tendencies for which the students with the greatest difficulties have not been the priority focus of attention.

Secondly, and closely linked to the above, the improvement in continuous teacher training entails not only advances in conditions and greater ease of access from the macro level or educational policies, but it is also incumbent on and challenges schools in the same way. If, taking up the recommendations of the OECD (2018), we consider as another of the keys to the improvement of the potential benefits derived from the capacity of autonomy of the schools for the selection and hiring of teachers, it is possible to establish a double aspect: In the first instance, the need for the schools to have a head team capable of sustaining pedagogical leadership that involves the entire educational community in a common and democratic project with the potential for the full integration in it of newly hired staff. The welcome plans for new teachers have been one of the initiatives pointed at to cultivate the conditions that foster a sense of belonging among teachers. Additionally, in the second instance, there is the need to promote the foundations so that the educational community can become capable of joining efforts and getting involved and collaborating in a common goal, with no other purpose than to achieve the success of all students with the highest expectations for every and each of the students. Therefore, within the democratic and inclusive approach to the training in schools as the axis of improvement and transformation and based on the contextual needs of each one of them and, especially, of their students, this is an issue that on the one hand is a priority but on the other does not seem easy or immediate. This since the case as it happens, among others reasons, due to the modification not only of more obvious aspects such as some of those already addressed, but also others

that are more difficult to perceive and are closely linked to the organizational culture of the schools that, in many cases, follows the evolution marked by the emblem of the dominant deficit approach (González and Cutanda 2020a, 2020b).

Thirdly, a final reflection is aimed at teachers themselves. The analysis of the data revealed the existence of highly internalized perceptions among teachers, opposing to what has been unleashed from theory and research on the importance of their training and that, curiously, only comes to light when they are directly involved in this type of programme. These are the reasonings by which it was justified that the necessary preparation to serve students at risk is a matter that concerns only certain specialists, who are the ones who have to be responsible for these programmes and their students. This is an issue that highlights the weight of such approaches when sustaining and shielding the dynamics of operation in the schools against the alterations that come with providing the precise response to the most vulnerable students, in which we could say that the teaching staff "accommodates". However, the results directly affect the teachers themselves, reducing their perception and motivation in many cases, and having as a counterpart a situation close to what could be described around the "figure of burnout in relation to teachers": high levels of stress, anxiety, frustration, greater work overload (if possible), isolation, etc. There is need for teachers who raise consciousness when working with every sort of student and, therefore, also with those students who are disengaging from school life together with their educational and training process. This is hardly feasible without minimal required training which transcends knowledge solely focused on areas or subjects. In this regard, it is worth noting the remodelling of initial training policies that is currently in full debate and discussion in the context of study (Document for debate. 24 proposals for the improvement in the teaching profession, January 2022, from the Spanish Ministry of Education and Vocational Training) that we hope will bear fruit along the lines that we have been exposing, and that will also have continuity with a high-level rethinking regarding the reflection on continuing training policies, their translation to the practice by the schools and, finally, in the awareness and commitment of the teaching staff.

In short, we appeal to a training thought, assumed and committed to by all (González and Cutanda 2015), for the sake of its incidence, not only in the learning of teachers but that can also transcend and have fruits in the highest levels of learning of each and every one of the students, understanding their achievements and successes in terms of equity and social justice.

**Author Contributions:** Conceptualization, M.T.C.-L. and M.B.A.-G.; methodology: M.T.C.-L.; software: M.T.C.-L. and M.B.A.-G.; validation, M.T.C.-L. and M.B.A.-G., formal analysis, M.T.C.-L. and M.B.A.-G.; investigation, M.T.C.-L. and M.B.A.-G.; resources: M.T.C.-L. and M.B.A.-G.; data curation, M.T.C.-L. and M.B.A.-G. writing—original draft preparation, M.T.C.-L. and M.B.A.-G.; writing—review and editing: M.T.C.-L. and M.B.A.-G.; visualization, M.T.C.-L.; supervision, M.T.C.-L. and M.B.A.-G.; project administration, M.T.C.-L.; funding acquisition M.T.C.-L. and M.B.A.-G. All authors have read and agreed to the published version of the manuscript.

**Funding:** This research was funded by: (1) Ministerio de Economía y Competitividad (Spain), Ph.D. Help Reference BES-2013-036943, linked to the doctoral thesis entitled "La implicación del alumnado absentista en su propio aprendizaje y el papel de la formación, el compromiso y el trabajo en el aula de los docentes. El caso de las Aulas Ocupacionales en la Región de Murcia" ("The involvement of absentee students in their own learning and the role of teachers' training, commitment and work in the classroom. The case of the Occupational Classrooms in the Region of Murcia"). (2) By the Ministerio de Innovación Ciencia y Universidades. Reference RTI2018-098806-B-I00: "Desarrollo Profesional Intergeneracional en Educación: Implicaciones en la Iniciación Profesional del Profesorado". Principal Researchers: Antonio Portela Pruaño y José Miguel Nieto Cano.

**Institutional Review Board Statement:** This doctoral thesis was reviewed and approved by the Consejería de Educación de la Comunidad Autónoma de la Región de Murcia (2016/11/28), and This research project was reviewed and approved by the Ethics Commission of the University of Murcia (approval identification code: 2087/2018).

**Informed Consent Statement:** Informed consent was obtained from all subjects involved in the study.

**Acknowledgments:** Our thanks to the participating students, teachers and to the organisations and individuals who have provided financial, technical and administrative support at both national and regional levels.

**Conflicts of Interest:** The authors declare no conflict of interest.

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
