# Peer review of "Teacher Training to Take Care of Students at Risk of Exclusion"

_socsci, doi:10.3390/socsci11120544_

Round 1
Reviewer 1 Report
Congratulations on the article! Please consider some comments I made below.
157-8: "in political, institutional, curricular and organizational and results keys" (consider revising "keys" - from political, institutional, curricular and organizational "perspectives")
190: the information collection (data)
194: from which the results that are analysed and presented here come (are extracted)
205: different groups of the same universe (?)
215: Research simple (?)
I'm not sure about the referencing - is it according to the journal's guidelines? I mean in particular the fact that you included the authors in brackets at the beginning (in the References section).
Author Response
Thank you very much for your considerations. we have proceeded to make the suggested changes.Reviewer 2 Report
I think that you should name the title Teacher training to take care of students at risk of exclusion
Author Response
Thank you very much for your considerations. we have proceeded to make the suggested changesReviewer 3 Report
Comments:
You should improve the coherence and cohession. You should avoid long sentences.
Line 21: in the Theoretical Framework you shouls include, at the beginning, more authors.
In addition, you can improve this part including actual literature.
Line 167: the conclusions drawn in the theoretical framework section should be supported by relevant citations.
Line 712: in conclusions, more contributions from the literature are missing and therefore more citations that support or contradict the results obtained in this study should be included.
Author Response
Thank you very much for your considerations. we have proceeded to make the suggested changes